# SFMBT2-Mediated Infiltration of Preadipocytes and TAMs in Prostate Cancer

**DOI:** 10.3390/cancers12092718

**Published:** 2020-09-22

**Authors:** Jungsug Gwak, Hayan Jeong, Kwanghyun Lee, Jee Yoon Shin, Taejun Sim, Jungtae Na, Jongchan Kim, Bong-Gun Ju

**Affiliations:** Department of Life Science, Sogang University, Seoul 04107, Korea; gwakjs79@sogang.ac.kr (J.G.); hayan90@sogang.ac.kr (H.J.); khistone@sogang.ac.kr (K.L.); osiris98@sogang.ac.kr (J.Y.S.); junely12@sogang.ac.kr (T.S.); pugokjebi@sogang.ac.kr (J.N.); jkimatsgu@sogang.ac.kr (J.K.)

**Keywords:** SFMBT2, prostate cancer metastasis, preadipocyte, TAMs, NF-κB, chemokine

## Abstract

**Simple Summary:**

The penetration of various cell types into the tumor microenvironment plays an important role in cancer progression, including metastasis. SFMBT2, an epigenetic factor, is downregulated in metastatic prostate cancer. The aim of the current study is to evaluate the role of SFMBT2 in regulating cell penetration into the prostate cancer microenvironment. Downregulation of SFMBT2 promotes infiltration of preadipocytes and TAMs by up-regulation of CXCL8, CCL2, CXCL10, and CCL20 expression. Expression of CXCL8, CCL2, CXCL10, and CCL20 is dependent on NF-κB activation in prostate cancer cells expressing low levels of SFMBT2. Moreover, increased IL-6 from infiltrated preadipocytes and TAMs further enhance migration and invasion of prostate cancer cells. Thus, SFMBT2 could be used as a novel biomarker and target for prostate cancer treatment.

**Abstract:**

Infiltration of diverse cell types into tumor microenvironment plays a critical role in cancer progression including metastasis. We previously reported that SFMBT2 (Scm-like with four mbt domains 2) regulates the expression of matrix metalloproteinases (MMPs) and migration and invasion of cancer cells in prostate cancer. Here we investigated whether the down-regulation of SFMBT2 regulates the infiltration of preadipocytes and tumor-associated macrophages (TAMs) in prostate cancer. We found that the down-regulation of SFMBT2 promotes the infiltration of preadipocytes and TAMs through up-regulation of CXCL8, CCL2, CXCL10, and CCL20 expression in prostate cancer. Expression of CXCL8, CCL2, CXCL10, and CCL20 was also elevated in prostate cancer patients having a higher Gleason score (≥8), which had substantially lower SFMBT2 expression. We also found that the up-regulation of CXCL8, CCL2, CXCL10, and CCL20 expression is dependent on NF-κB activation in prostate cancer cells expressing a low level of SFMBT2. Moreover, increased IL-6 from infiltrated preadipocytes and TAMs promoted migration and invasion of prostate cancer cells expressing a low level of SFMBT2. Our study may suggest that SFMBT2 a critical regulator for the infiltration of preadipocytes and TAMs into the prostate tumor microenvironment. Thus, the regulation of SFMBT2 may provide a new therapeutic strategy to inhibit prostate cancer metastasis, and SFMBT2 could be used as a potential biomarker in prostate cancer metastasis.

## 1. Introduction

Prostate cancer is the second most commonly diagnosed cancer in men over the age of 65 in the developed world. Although the 5-year survival rate is 100% for men with local or regional prostate cancer, it decreases to 30% in men with metastatic prostate cancer [1]. Prostate-specific antigen (PSA) is a widely used biomarker for the diagnosis of prostate cancer at an early stage. However, more sensitive and specific biomarkers such as prostate cancer gene 3 (PCA3) have been developed because of the limited sensitivity and specificity of PSA testing [2].

Tumor microenvironment surrounding cancer cells is mainly composed of normal and nonreactive stromal cells including fibroblasts, myofibroblasts, neuroendocrine cells, adipose cells, and immune cells [3]. Cancer cells change their microenvironment by extracellular matrix (ECM) remodeling and the secretion of cytokines, chemokines, and growth factors in either or both an autocrine and paracrine manner [4]. Recently, it has been suggested that obesity or adipose tissues play a critical role in prostate cancer progression such as metastasis and recurrence [5,6]. Periprostatic adipose tissue (PPAT) surrounding the prostate gland is regarded as an important prostate cancer microenvironment and it may be associated with a higher grade and aggressiveness of prostate cancer [7,8]. In patients with poorer prognostic outcomes, cancer cells invade into PPAT and contact with adipose tissue, which is defined as an extracapsular extension [9]. Thus, extracapsular extension leads to cross-talk between cancer cells and adipocytes, preadipocytes, immune cells, and fibroblasts in PPAT. In particular, adipocytes are reprogrammed by cancer cells to less differentiated adipocytes, promoting aggressive and metastatic prostate cancer [10]. In addition, adipokines and high energy fatty acids secreted by adipocytes influence prostate cancer progression [9,11,12,13,14]. Tumor-associated macrophages (TAMs) also are present in the tumor microenvironment and affect prostate cancer progressions such as angiogenesis, metastasis, immune suppression, and drug resistance [15,16]. Depending on the signals present in the tumor microenvironment, macrophages polarize into M1 and M2 subtypes. Although M1 macrophages (classically activated) produce TNF-α, IFNγ, IL-12, and IL-23 and have pro-inflammation functions, M2 TAMs contribute to immune suppression, tumor growth, and metastasis via secretion of IL1β, IL-6, CXCL8, and VEGF [16,17,18,19].

Accumulating evidence demonstrates that polycomb group proteins (PcGs) play an important role in prostate cancer progression. For instance, polycomb repressive complex 1 (PRC1) promotes metastasis of double negative [androgen receptor (AR) and neuroendocrine] prostate cancers by CCL2 expression, which regulates TAM self-renewal and recruitment [20]. In castration-resistant prostate cancer, histone H3K27 methyltransferase EZH2, a component of PRC2, is upregulated and methylates AR, resulting in regulation of AR target gene expression [21,22]. EZH2 also suppresses the tumor suppressor PSP94 gene via trimethylation of histone H3K27 at the promoter [23]. Recently, we found that Scm-like with four MBT domains protein 2 (SFMBT2), another PcG protein [24,25], is involved in prostate cancer metastasis [26]. SFMBT2 interacts with YY1 and repressive histone marks to repress gene expression of matrix metalloproteinases (MMPs), which are critical for cancer cell migration and invasion. The expression level of SFMBT2 is low in highly metastatic prostate cancer cells. In addition, patients having a higher Gleason score (≥8) had substantially lower SFMBT2 expression than patients with a lower Gleason score [26].

In this study, we investigated whether SFMBT2 regulates cell infiltration into the prostate cancer microenvironment. We found that the down-regulation of SFMBT2 promotes the up-regulation of CXCL8, CCL2, CXCL10, and CCL20 expression in prostate cancer cells, resulting in elevated infiltration of preadipocytes and TAMs. Up-regulation of CXCL8, CCL2, CXCL10, and CCL20 is dependent on NF-κB activation. In addition, IL-6 from infiltrated preadipocytes and TAMs further promotes migration and invasion of prostate cancer cells expressing a low level of SFMBT2.

## 2. Results

### 2.1. SFMBT2 Regulates Expression of Chemokines

We previously reported that down-regulation of SFMBT2 promotes prostate cancer metastasis through the up-regulation of MMPs and that expression level of SFMBT2 inversely correlates with the prognosis of prostate cancer patients such as invasion and metastasis [26]. Given that infiltration of adipocytes or macrophages into the tumor microenvironment contributes to prostate cancer progression [27,28], we further investigated whether SFMBT2 regulates the expression of chemokines, which are linked to cell infiltration. Because poorly metastatic LNCaP cells express a relatively high level of SFMBT2 compared with highly metastatic PC3 and DU145 cells and because LNCaP cells expressing a low expression level of SFMBT2 by RNA interference showed enhanced migration and invasion [26], we used mainly LNCaP cells as a cell model in this study.

The culture medium from LNCaP cells stably transfected with control or SFMBT2 shRNA was incubated with a cytokine antibody array membrane. In comparison with LNCaP cells stably transfected with control shRNA, up-regulation of CD147, CXCL8, CXCL10, LCN2, CCL2, CCL20, PDGF AA, VEGF, and Ang-2 expression was observed in LNCaP cells stably transfected with SFMBT2 shRNA (Figure 1A). Among these chemokines and cytokines, we selected CXCL8, CCL2, CXCL10, and CCL20 for further study because of their critical role in cell infiltration into the tumor microenvironment [27,29,30,31,32]. We confirmed increased expression of CXCL8, CCL2, CXCL10, and CCL20 in LNCaP cells stably transfected with SFMBT2 shRNA by quantitative RT-PCR (Figure 1B). To examine the expression of chemokines in prostate cancer tissues, LNCaP cells stably transfected with SFMBT2 shRNA were injected intraprostatically in nude mice [26]. We observed the increased expression of CXCL8, CCL2, CXCL10, and CCL20 in prostate cancer tissues induced by intraprostatic injection of LNCaP cells stably transfected with SFMBT2 shRNA compared with prostate tissues injected with LNCaP cells stably transfected with control shRNA (Figure 1C). Immunohistochemical analysis of human normal and prostate cancer tissue array consistently demonstrated the increased expression of CXCL8, CCL2, CXCL10, and CCL20 in prostate cancer tissues (Figure 1D). We further analyzed these data to investigate the relationship between the expression level of chemokines and Gleason scores (See Materials and Methods). Expression levels of CXCL8, CCL2, CXCL10, and CCL20 in prostate cancer with Gleason scores of ≥8 were higher than those with Gleason scores of ≤7 (Figure 1E). We also reported that the expression level of SFMBT2 inversely correlates with Gleason scores [26]. Thus, expression levels of chemokines and SFMBT2 were analyzed in prostate cancer tissues. Our analysis indicated the high expression of CXCL8, CCL2, CXCL10, and CCL20 in prostate cancer tissues expressing a low expression level of SFMBT2, which is correlated with high Gleason scores (Figure 1F, Appendix A) [26].

### 2.2. Expression Level of SFMBT2 Inversely Correlates with Infiltration of Preadipocytes and TAMs

We next investigated whether SFMBT2 regulates the infiltration of adipocytes and TAMs by up-regulation of chemokine expression in prostate cancer. Because there were no obvious morphological and biochemical characteristics of adipocytes in prostate cancer tissues induced by intraprostatic injection of LNCaP cells stably transfected with SFMBT2 shRNA (data not shown), we decided to examine infiltration of preadipocytes, which may play an important role in prostate and breast cancer metastasis [33,34,35,36]. We first examined the expression of preadipocyte markers such as Pref-1 and CD29 [33,37]. Immunohistochemical analysis indicated increased numbers of Pref-1 and CD29 positive cells in prostate cancer induced by intraprostatic injection of LNCaP cells stably transfected with SFMBT2 shRNA (Figure 2A). We consistently found that the number of Pref-1 or CD29 positive cells is increased in prostate cancer tissues compared with normal prostate tissues using human normal and prostate cancer tissue array (Figure 2B,C). As with the expression level of CXCL8, CCL2, CXCL10, and CCL20, the number of Pref-1 or CD29 positive cells was increased in prostate cancer tissues with high Gleason scores of ≥8 (Figure 2C). In addition, the SFMBT2 expression level was inversely related to the number of Pref-1 or CD29 positive cells (Figure 2C).

Chemokines play an important role in the infiltration of TAMs [38]. We further examined the expression of TAM-specific cell surface markers such as CD163 and CD206 [39]. The number of CD163 and CD206 positive cells was increased in prostate cancer tissues induced by intraprostatic injection of LNCaP cells stably transfected with SFMBT2 shRNA and in prostate cancer tissues of patients (Figure 2D,E). The number of CD163 and CD206 positive cells was proportionally related to Gleason scores and inversely related to the expression level of SFMBT2 (Figure 2F). These results suggest that prostate cancer cells expressing a low level of SFMBT2 may promote the infiltration of preadipocytes and TAMs by up-regulation of chemokines including CXCL8, CCL2, CXCL10, and CCL20.

### 2.3. Down-Regulation of SFMBT2 Induces Migration of Preadipocyte and TAMs

We further tested the function of SFMBT2 in cell migration using 3T3-L1 preadipocytes and TAMs polarized from Raw264.7 cells. Transwell assay showed that culture media from LNCaP cells stably transfected with SFMBT2 shRNA promote enhanced migration of 3T3-L1 cells (Figure 3A and Appendix A). However, the addition of antibodies against CXCL8, CCL2, CXCL10, and CCL20 and a mixture of antibodies into the culture media from LNCaP cells stably transfected with SFMBT2 shRNA suppressed 3T3-L1 cell migration (Figure 3A and Appendix A). We consistently found that addition of CXCL8, CCL2, CXCL10, and CCL20 into 3T3-L1 cell culture media (RPMI 1640) increases migration of 3T3-L1 cells (Figure 3B and Appendix A). Antibody treatment abolished the chemokine-induced migration of 3T3-L1 cells (Figure 3B and Appendix A). We next examined the expression of receptors for chemokines (CCR2, CCR4, CCR6, CXCR1, CXCR2, CXCR3) in 3T3-L1 cells. The expression of chemokine receptors was increased in 3T3-L1 cells incubated with culture media from LNCaP cells stably transfected with SFMBT2 shRNA (Figure 3C). Similar to 3T3-L1 preadipocytes, we observed chemokine-dependent migration of TAMs (Figure 3D,E, Appendix A). In addition, the expression of chemokine receptors (CCR2, CCR6, CXCR1, CXCR2, CXCR3) was increased in TAMs incubated with culture media from LNCaP cells stably transfected with SFMBT2 shRNA (Figure 3F). We also tested whether culture media from PC3 cells, highly metastatic prostate cancer cells endogenously expressing a low level of SFMBT2, have a similar effect on the migration of 3T3-L1 preadipocytes and TAMs [26]. As expected, culture media from PC3 cells promoted the migration of 3T3-L1 preadipocytes and TAMs (Appendix A). In contrast, over-expression of SFMBT2 suppressed the up-regulation of CXCL8, CCL2, CXCL10, and CCL20 expression in PC3 cells (Appendix A). Culture media from PC3 cells over-expressing SFMBT2 suppressed migration of 3T3-L1 preadipocytes and TAMs (Appendix A).

Using a tumor xenograft model, we further confirmed SFMBT2-mediated infiltration of preadipocytes and TAMs. LNCaP cells stably transfected with SFMBT2 shRNA were implanted subcutaneously into the flank of nude mice. After 4 weeks, RFP-expressing 3T3-L1 cells were injected into the tail vein of the same mouse (Figure 4A). Immunohistochemistry using anti-RFP antibodies revealed that the number of infiltrated 3T3-L1 cells was increased in tumors originated from LNCaP cells stably transfected with SFMBT2 shRNA (Figure 4B). In addition, RFP-expressing 3T3-L1 cells were positive for Pref-1 and CD29 (Figure 4C). We also found an increased infiltration of TAMs immunostained with anti-CD163 and anti-CD206 antibodies in tumor originated from LNCaP cells stably transfected with SFMBT2 shRNA (Figure 4D). RNA analysis further confirmed the increased expression of the marker genes for preadipocytes (Pref-1, CD24, CD29, Sca-1) and TAMs (CD163, CD206) in tumors originated from LNCaP cells stably transfected with SFMBT2 shRNA (Appendix A).

### 2.4. NF-κB Up-Regulates CXCL8, CCL2, CXCL10, and CCL20 Gene Expression in SFMBT2 Knockdown LNCaP Cells

To investigate the molecular mechanism underlying down-regulation of SFMBT2-mediated up-regulation of CXCL8, CCL2, CXCL10, and CCL20 expression, we first performed ChIP assay. Unexpectedly, SFMBT2 was not enriched on the promoters of CXCL8, CCL2, CXCL10, and CCL20 genes in LNCaP cells (Appendix A). Instead, NF-κB (p65 and p50) was enriched on the promoters with acetylated H3 and p300 coactivator in LNCaP cells stably transfected with shSFMBT2 (Figure 5A). BAY 11–7085 treatment, which inhibits IκBα phosphorylation, abolished up-regulation of CXCL8, CCL2, CXCL10, and CCL20 gene expression in LNCaP cells stably transfected with SFMBT2 shRNA (Figure 5B). The promoter assay also demonstrated that SFMBT2 knockdown activates NF-κB signaling (Figure 5C). Consistent with our previous report [26], we confirmed increased phosphorylation and degradation of IκB as well as nuclear translocation of NF-κB p65 in LNCaP cells stably transfected with SFMBT2 shRNA (Figure 5D,E). These results indicate that down-regulation of SFMBT2 activates the NF-κB signaling in prostate cancer cells.

### 2.5. Preadipocytes and TAMs Regulate Migration and Invasion of Prostate Cancer Cells by Up-Regulation of IL-6 Expression

Given that reciprocal interaction between infiltrated cells and prostate cancer cells plays a critical role in metastasis [27,28], we further investigated the effects of infiltrated preadipocytes and TAMs on migration and invasion of prostate cancer cells. LNCaP cells stably transfected with control or SFMBT2 shRNA were incubated with culture medium from 3T3-L1 preadipocytes or TAMs. We found that the number of migrating and invading LNCaP cells stably transfected with SFMBT2 shRNA was increased compared with LNCaP cells stably transfected with control shRNA (Figure 6A–D and Appendix A). It is well known that IL-6 promotes cancer cell growth, invasion, migration, and epithelial-mesenchymal transition (EMT), resulting in prostate metastasis [40,41]. Thus, we tested whether IL-6 secreted from preadipocytes and TAMs affects migration and invasion of prostate cancer cells. As expected, the addition of anti-IL-6 antibody inhibited migration and invasion of LNCaP cells stably transfected with SFMBT2 shRNA in culture media from 3T3-L1 cells or TAMs (Figure 6A–D and Appendix A). Moreover, incubation of culture media from LNCaP cells stably transfected with SFMBT2 shRNA up-regulated IL-6 expression in 3T3-L1 cells and TAMs (Figure 6E,F). Immunohistochemical analysis demonstrated that the expression of IL-6 is increased in prostate cancer induced by intraprostatic injection of LNCaP cells stably transfected with SFMBT2 shRNA (Figure 6G). Consistently, we found increased IL-6 in prostate cancer tissues of patients compared with normal prostate tissues (Figure 6H). The expression of IL-6 was proportionally related to Gleason scores and inversely related to SFMBT2 expression (Figure 6I).

Taken together, our results suggest that down-regulation of SFMBT2, which is correlated with a high Gleason score (≥8), promotes the infiltration of preadipocytes and TAMs through the up-regulation of chemokine expression in prostate cancer cells. Furthermore, IL-6 from infiltrated preadipocytes and TAMs promotes migration and invasion of prostate cancer cells (Figure 7).

## 3. Discussion

We previously demonstrated that down-regulation of mammalian PcG protein SFMBT2 promotes prostate cancer metastasis through the up-regulation of MMPs [26]. Prostate cancer patients with higher Gleason scores (≥8) have substantially lower SFMBT2 expression than patients with lower Gleason scores, indicating that SFMBT2 may have an anti-metastatic function [26].

In this study, we investigated whether SFMBT2 regulates cell infiltration into the prostate tumor microenvironment, which has a critical role in metastasis. SFMBT2 knockdown resulted in increased expression of CXCL8, CCL2, CXCL10, and CCL20 in LNCaP cells. In addition, expression of CXCL8, CCL2, CXCL10, and CCL20 was also increased in prostate cancer tissue induced by intraprostatic injection of LNCaP cells stably transfected with SFMBT2 shRNA and in the tissue of prostate cancer patients. Also, the expression level of CXCL8, CCL2, CXCL10, and CCL20 was proportionally related to a high Gleason score of ≥8, which seems closely related to prostate cancer invasion and metastasis [42,43,44]. Their expression was also increased in prostate cancers expressing a low level of SFMBT2, which are the more invasive cancers [26]. In support of our findings, it has been demonstrated that CXCL8 (also known as IL-8) contributes to prostate cancer progressions such as tumorigenesis, metastasis, and chemoresistance [45,46]. High serum level and expression of CXCL8 were correlated with an advanced pathological stage, high Gleason score, metastasis, and recurrence [47]. CCL2 [also known as MCP1 (monocyte chemoattractant protein 1)] is highly expressed in prostate cancer cells with high metastatic potential [48]. CCL2 is also reported as a key chemokine for bone metastasis and cancer drug resistance in prostate cancer [49,50]. CXCL10 (also known as IP10) promotes cell motility and invasiveness in both DU-145 and PC-3 cells via PLCβ3 and μ-calpain activation [51]. Over-expression of CCL20 has been found in prostate cancer and it promotes tumor growth [52].

Increased expression of chemokines led us to investigate the infiltration of several cell types such as adipocytes and macrophages into the prostate cancer microenvironment [53,54,55]. Because there were no morphological and biochemical characteristics of adipocytes in prostate cancer tissue induced by intraprostatic injection of LNCaP cells stably transfected with SFMBT2 shRNA, we tested whether preadipocytes, which are mesenchymal adipocyte progenitor cells, are present in the tumor. Although the exact contribution of preadipocytes to cancer is not clear, it has been shown recently that preadipocytes are an important component of cancer progression in breast and prostate cancer [33,34,35,36]. We found increased infiltration of Pref-1 and CD29 positive preadipocytes [33,56,57,58] in prostate cancer tissues induced by intraprostatic injection of LNCaP cells stably transfected with SFMBT2 shRNA and in tissues of prostate cancer patients. Infiltration of Pref-1 and CD29 positive preadipocytes was proportionally related to a high Gleason score of ≥8 and inversely related to the SFMBT2 level. A previous report demonstrated consistently that increased infiltration of Pref-1 and CD29 positive preadipocytes promote prostate cancer metastasis via the mR301a/AR/TGFβ1/Smad/MMP9 pathway [33]. Tumor-induced by transplantation of PC3 prostate cancer cells recruits adipose tissue-derived stromal/stem cells [59]. In addition, CXCL8 secreted from prostate cancer cells promotes the recruitment of adipose stromal cells from white adipose tissue (WAT), and infiltrated adipocytes affect prostate cancer progression [27]. By using an in vitro cell model, it has been demonstrated that CCL2 and CCL20 promote the migration of adipose stem cells [60,61].

Although we could not provide the origin of preadipocytes, accumulating evidence indicates a critical role of periprostatic adipose tissue (PPAT) surrounding the prostate gland in prostate cancer progression [7,11]. A higher fat ratio of PPAT is associated with a higher Gleason score [62]. Fluorescence-activated cell sorting (FACS) analysis shows that a greater amount of adipose stem cells is present in PPAT compared with visceral adipose tissue, and the amount of adipose stem cells is increased in prostate cancer [9]. In addition, increased angiogenesis and arteriolar size are found in the PPAT of prostate cancer patients, suggesting that PPAT may be the origin of preadipocytes [7]. However, we cannot exclude the possibility that preadipocytes are present in prostate cancer tissue or that there is the infiltration of preadipocytes from visceral adipose tissue [9,63,64]. In prostate cancer, infiltrated preadipocytes increase the invasion of prostate cancer cells. For instance, the co-culture of preadipocytes suppresses AR expression through the up-regulation of miR-301a expression in prostate cancer cells, resulting in the up-regulation of TGFβ1, Smad, and MMP9 expression for metastasis [33]. Similarly, incubation of androgen-independent prostate cancer cells such as RM1 with culture media from 3T3-L1 cells increases migration and invasion of prostate cancer cells [34]. In breast cancer, exosome containing miR-140 or IL-6 from preadipocytes promotes breast cancer progression [35,36].

In the tumor microenvironment, the increased number of infiltrated TAMs is found in higher-grade prostate cancer and is associated with worse prognosis [65,66,67]. Consistent with our results, prostate cancer-derived CXCL8 induces chemotaxis of macrophage-like THP-1 cells [68]. Up-regulation of CCL2 induced by metastatic prostate cancer cells promotes the infiltration of TAMs [20,69]. In addition, increased expression of CCL2 promotes prostate cancer growth and metastasis through TAM infiltration [52,70,71]. Although there is no direct evidence supporting CXCL10 and CCL20-mediated TAM infiltration in prostate cancer, CXCL10 and CCL20 promote TAM recruitment in colorectal and breast cancer, respectively [31,72].

We previously reported that SFMBT2 acts as a transcriptional repressor for MMPs in prostate cancer cells [26]. In contrast, no enrichment of SFMBT2 at the gene promoters of CXCL8, CCL2, CXCL10, and CCL20 was observed in LNCaP cells. Instead, our ChIP data suggest that NF-κB activation leads to the up-regulation of CXCL8, CCL2, CXCL10, and CCL20 gene expression in SFMBT2 knockdown LNCaP cells. We cannot address the exact molecular mechanism underlying SFMBT2-dependent NF-κB regulation in this study. However, it has been previously shown that NF-κB activation may be associated with prostate cancer progression. For example, NF-κB is more activated in highly metastatic prostate cancer cells than in poorly metastatic cells [73,74,75]. The nuclear localization of NF-κB was increased in prostate cancer with high Gleason scores and metastatic prostate cancer [76,77]. Thus, inhibition of NF-κB results in the decreased invasion, angiogenesis, and metastasis through down-regulation of VEGF, CXCL8, and MMP-9 gene expression [78]. Abnormal NF-κB activation may be caused by high levels of cytokines, microRNAs, genetic alternation of NF-κB itself or genes related to signaling, and over-activation of signaling in prostate cancer [79,80,81,82,83]. Similarly, we found that SFMBT2 acts as a transcriptional repressor of TNFα gene expression and knockdown of SFMBT2 results in increased expression of TNFα, which activates the NF-κB signaling and plays an important role in prostate cancer progression (Appendix A) [84,85,86]. In addition, HoxB13, which is identified as a target gene of SFMBT2 transcriptional repressor [87], may contribute to NF-κB-mediated prostate cancer metastasis because over-expression of HoxB13 leads to reduced expression of IκBα and enhanced the nuclear translocation of NF-κB p65 in LNCaP cells [88].

It is known that cells infiltrated into the tumor microenvironment critically contribute to prostate cancer progression via secretion of various paracrine factors such as cytokines and growth factors [7,89]. In this study, we found that IL-6 derived from 3T3-L1 preadipocytes and TAMs promotes migration and invasion of LNCaP cells. IL-6 mediated cell migration and invasion were enhanced in SFMBT2 knockdown LNCaP cells compared with control shRNA-transfected LNCaP cells. In addition, IL-6 expression was proportionally related to a high Gleason score of ≥8 and increased in prostate cancers expressing a low level of SFMBT2. Consistent with our findings, high levels of IL-6 were detected in prostate cancer tissues and sera of patients with advanced prostate cancer [90,91,92,93,94]. In addition, IL-6 promotes prostate cancer cell growth, invasion, migration, and epithelial-mesenchymal transition (EMT), resulting in metastasis [95,96,97].

In this study, we suggest that chemokines produced from prostate cancer cells expressing a low level of SFMBT2 promote the infiltration of preadipocytes and TAMs. These chemokines also affect diverse cell types, contributing to prostate cancer progression. For example, CXCL8 promotes tumor progression by increased infiltration of polymorphonuclear myeloid-derived suppressor cells (PMN-MDSCs) [98]. Over-expression of CCL2 increases tumor growth and bone metastasis through the recruitment of macrophages and osteoclast to tumor sites [49]. However, roles of CXCL10 and CCL20 in different cell types are not fully understood in prostate cancer at present. Further investigation is needed to explore the exact role of chemokines induced by down-regulation of SFMBT2 in prostate cancer progression.

In conclusion, our study provides evidence that the down-regulation of SFMBT2 promotes prostate cancer metastasis by the recruitment of preadipocytes and TAMs through NF-κB-dependent up-regulation of chemokine expression (Figure 7). In turn, IL-6 from infiltrated preadipocytes and TAMs further promotes migration and invasion of prostate cancer cells. Thus, the regulation of SFMBT2 may provide a new therapeutic strategy to inhibit prostate cancer metastasis, and SFMBT2 could be used as a potential biomarker in prostate cancer metastasis.

## 4. Materials and Methods

### 4.1. Cell Lines

LNCaP, Raw264.7, PC3, and 3T3-L1 cells were purchased from the American Type Culture Collection (ATCC, Manassas, VI, USA) and cultured in RPMI 1640 supplemented with 10% fetal bovine serum, 100 mg/mL penicillin, and 100 mg/mL streptomycin. Cell lines were cultured in an incubator with 5% CO2 at 37 °C. For stable transfection of SFMBT2, LNCaP cells were transfected with GIPZ lentiviral SFMBT2 shRNA containing GFP (RHS4531, GE Dharmacon, Lafayette, CO, USA). Stable clones were established by culturing LNCaP cells in media containing 1 µg/mL puromycin [26]. GIPZ lentiviral control shRNA (RHS4346, GE Dharmacon) was used as a control. To obtain 3T3-L1 cells expressing red fluorescent protein (RFP), cells were transfected with pCS4+ RFP expressing vector and stable clones were selected by treatment of 400 µg/mL G418. For TAMs polarization, Raw264.7 cells were treated with IL-4 (20 ng/mL) and IL-13 (20 ng/mL) for 2 days [99,100].

### 4.2. Cytokine Array

Chemokine and cytokine profiles were analyzed using a Proteome Profiler Human XL Cytokine Array Kit (ARY022B, R&D Systems, Minneapolis, MN, USA) according to the manufacturer’s instructions.

### 4.3. RNA Extraction and Quantitative RT-PCR

Total RNA was extracted using Trizol. One microgram of total RNA was used in the reverse transcription reaction using PrimeScript RT Master Mix (RR036, Takara, Kyoto, Japan). Quantitative real-time RT-PCR was performed using qPCR 2X PreMIX (RT500S, Enzynomics, Daejeon, Korea) with an Mx3000p qPCR machine (Agilent Technologies, Santa Clara, CA, USA). PCR conditions were 30 s at 95 °C, 40 cycles of 95 °C for 5 s, and 60 °C for 34 s. Expression was calculated from the cycle threshold (Ct) value using the ΔCt method for quantification. GAPDH mRNA level was as used for normalization. Oligonucleotide primers of real-time PCR are described in Appendix A.

### 4.4. Promoter Assay

LNCaP cells were transfected with NF-κB firefly luciferase reporter containing NF-κB binding sites [101], pCMV-Renilla luciferase, or SFMBT2 shRNA using Lipofectamine 2000 reagent (Invitrogen, Carlsbad, CA, USA) and incubated for 48 h. LNCaP cells were co-transfected with TNFα promoter-driven firefly luciferase reporter, pCMV-Renilla luciferase, or SFMBT2 expressing plasmid using Lipofectamine 2000 reagent. Cells were washed twice, suspended in 100 μL of Passive lysis buffer, and then luciferase assay was performed using the Dual-Luciferase Reporter Assay System (E1960, Promega, Madison, WI, USA) with a Lumat BL 9507 luminometer (Berthold Technologies, Bad Wildbad, Germany). Firefly luciferase activity was normalized to Renilla luciferase activity.

### 4.5. Western Blot Analysis

Cells were lysed on ice with RIPA buffer [20 mM Tris-HCl (pH 7.5), 150 mM NaCl, 1% NP-40, 0.5% sodium deoxycholate, and 0.1% SDS] containing protease inhibitors. Protein concentration was determined by the Bradford assay. Denatured proteins were separated by SDS-PAGE and transferred to PVDF membranes. The membrane was blocked in PBS containing 5% non-fat milk, washed 3 times with PBST (0.1% Tween-20), and incubated with primary antibodies overnight at 4 °C. Primary antibodies are described in Appendix A.

### 4.6. ELISA

TNF-α in culture media was measured using Human TNF-α ELISA Kit (E-80TNF, KOMA BIOTECH) according to the manufacturer’s protocol.

### 4.7. Immunocytochemistry

Cells were fixed for 10 min with 1% paraformaldehyde in PBS at room temperature, washed with PBS, and permeabilized with PBST solution (0.5% Triton X-100 in PBS) for 30 min. The cells were blocked with 5% BSA in PBST solution for 1h at room temperature and incubated with primary antibody for overnight at 4 °C. After the slides were rinsed three times in PBS, cells were incubated with the secondary antibodies conjugated to TRITC for 1h. Nuclei were identified using DAPI staining and images were acquired with a confocal microscope (Leica TCS SPE, Mannheim, Germany). Primary antibodies are described in Appendix A.

### 4.8. Cell Migration and Invasion Assay

Cell migration assays were performed using modified Boyden Chambers (3422, Corning Costar). For invasion assays, cells were allowed to invade through a matrigel-coated membrane (354480, Corning Costar). Cells were added into the upper chamber of a 24-well plate (2.5 × 10^4^ cells) and incubated with media containing either recombinant chemokine (100 ng) or antibodies (1 μg) or both for 24 h (Appendix A). Transwell membranes were removed, and cells were stained for 10 min in 0.3% neutral red.

### 4.9. In Vivo Metastasis Assay

All animal procedures in this study were approved by the committee for experimental animal research at Sogang University (IACUCSGU2015_01). Male athymic BALB/c nude mice (5 weeks old, 21 g of average body weight; DBL, Korea) were used with two biological repeats (n = 3/group). For intraprostatic injection [102], mice were anesthetized with an intraperitoneal injection of 2, 2, 2-tribromoethanol (0.24 μg/g of body weight) and placed in a supine position. A midline incision was made in the lower abdomen and the prostate was exteriorized. One million LNCaP cells stably transfected with shContol-GFP or shSFMBT2-GFP in 50 μL PBS were injected into the dorsolateral side of the prostate. The incision was closed with sutures. At 5 weeks after injection, mice were dissected, and organs were removed.

### 4.10. Xenograft

All animal procedures in this study were approved by the committee for experimental animal research at Sogang University (IACUCSGU2015_01). Male athymic BALB/c nude mice (5 weeks old, 21 g of average body weight) were used with two biological repeats (n = 3/group). One million LNCaP cells stably transfected with shControl or shSFMBT2 were subcutaneously implanted into the left flank. Tumor size was measured with a caliper and volume was calculated as length × width; 2 × 0.5 cm. At 1 week after injection, 3T3-L1 cells expressing RFP were injected into the tail vein. After 1 week, tumors were recovered and processed for further studies.

### 4.11. Immunohistochemistry

Human normal and prostate cancer tissue arrays (2 mm core diameter) with patient information such as Gleason score were purchased from Super Bio Chip Laboratories (Korea) (Appendix A). Mouse tumors were fixed in formalin, embedded in paraffin, and cut into 4 μm thick sections. The sections were deparaffinized in xylene and rehydrated in a graded series of ethanol. For heat-mediated antigen retrieval, the tissue slides were incubated in citrate buffer (pH 6.0) at 96 °C for 20 min. After blocking with 1% BSA in PBS, slides were incubated with primary antibodies overnight at 4 °C. Then, slides were washed with PBS containing 0.025% Tween 20 (PBST) and incubated with secondary antibodies for 1h at room temperature. Nuclei were identified using DAPI staining. Images were acquired with a confocal microscope (Leica TCS SPE, Mannheim, Germany). Intensity values of images from whole tissue were analyzed with ImageJ software (Version 1.49, NIH, Bethesda, MD, USA).

The image values for overall intensity was based on a 4-point system: 0, 1, 2, and 3 (for none, light, medium, or dark staining) and the percentage of positively stained cells was based on a 5-point system: 0% = 0, 1–10% = 1, 11–50% = 2, 51~80% = 3, 81–100% = 4. The final score was determined by multiplying the intensity scores with the scores of positively stained cells. Thus, the final scores were in a range from 0 to 12. A score of 0 to 4 was called “low”, a score of 5 to 8 was called “moderate”, and a score of 8~12 was called “high” [26]. Scoring was done by two reviewers who were blind to the results. *p* values were determined by a Kruskal–Wallis test and the values were expressed as means ± S.D. (n = 3). The number of CD29, Pref-1, CD206, or CD163-positive cells was counted. The results were the number of positive cells as the mean ± S.D. Primary antibodies are described in Appendix A.

### 4.12. Chromatin Immunoprecipitation (ChIP)

ChIP was performed according to the method previously described [26]. Normal rabbit IgG (sc-2027, Santa Cruz Biotechnology, Santa Cruz, CA, USA) or mouse IgG (sc-2025, Santa Cruz Biotechnology) was used as a negative control. Quantitative PCR of isolated DNA fragments was performed using the primers listed in Appendix A. The relative proportions of immunoprecipitated fragments were determined using the ΔCt comparative method based on the threshold cycle (Ct) value for each PCR reaction and normalized to input genomic DNA.

### 4.13. Statistical Analyses

Results are presented as the means ± standard deviation. Kruskal–Wallis test was performed using statistical software GraphPad Prism 8 (Version 6, GraphPad Software, San Diego, CA, USA). The difference was considered statistically significant at *p* < 0.05.

## 5. Conclusions

Down-regulation of SFMBT2, which is found in patients with a higher Gleason score (≥8), promotes the infiltration of preadipocytes and TAMs by up-regulation of chemokine expression. Infiltrated preadipocytes and TAMs further enhance migration and invasion of prostate cancer cells. SFMBT2 could be used as a novel biomarker and target for prostate cancer treatment.

## Figures and Tables

**Figure 1 cancers-12-02718-f001:**
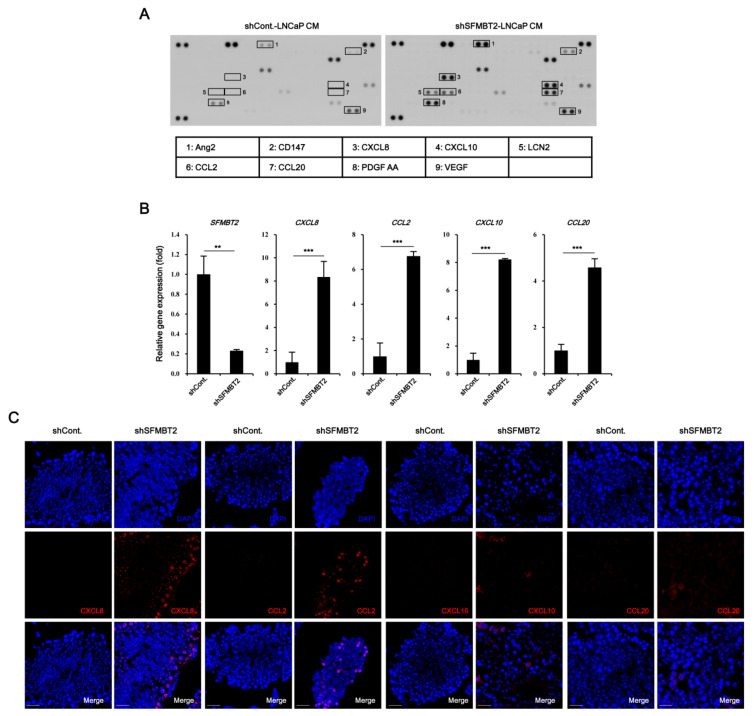
Down-regulation of SFMBT2 up-regulates the expression of chemokine genes in prostate cancer. (**A**) Increased expression of chemokines in LNCaP cells stably transfected with SFMBT2 shRNA. Culture media (CM) from LNCaP cells stably transfected with control (shCont) or SFMBT2 shRNA (shSFMBT2) were incubated with cytokine antibody array membrane. (**B**) Increased expression of CXCL8, CCL2, CXCL10, and CCL20 genes in LNCaP cells stably transfected with SFMBT2 shRNA. RNA was analyzed in LNCaP cells stably transfected with control or SFMBT2 shRNA. (**C**) Increased expression of CXCL8, CCL2, CXCL10, and CCL20 in prostate cancer induced by intraprostatic injection of LNCaP cells stably transfected with SFMBT2 shRNA. LNCaP cells (1 × 10^6^ cells) transfected stably with control or SFMBT2 shRNA were injected into the prostate (dorsal lobe) of nude mice (n = 3/group). At week 5 post-injection, the prostate was harvested, fixed, sectioned, and immunostained with anti-CXCL8, anti-CCL2, anti-CXCL10, and anti-CCL20 antibodies. Representative images are shown. Nuclei were identified using DAPI staining. Scale bar, 25 μm. Expression was quantified using the ImageJ program. (**D**) Increased expression of CXCL8, CCL2, CXCL10, and CCL20 in prostate cancer tissues of patients. Immunohistochemical staining of a tissue array from prostate cancer patients was performed using anti-CXCL8, anti-CCL2, anti-CXCL10, and anti-CCL20 antibodies. Representative images are shown. Nuclei were identified using DAPI staining. Scale bar, 25 μm. Expression was quantified using the ImageJ program. (**E**) Proportional relationship of Gleason scores (GS) with the expression level of CXCL8, CCL2, CXCL10, and CCL20 in prostate cancer tissues of patients. (**F**) Inverse relationship of SFMBT2 expression with the expression level of CXCL8, CCL2, CXCL10, and CCL20 in prostate cancer tissues of patients. All data represent mean ± S.E.M. Significance values were ** *p* ≤ 0.01 and *** *p* ≤ 0.005.

**Figure 2 cancers-12-02718-f002:**
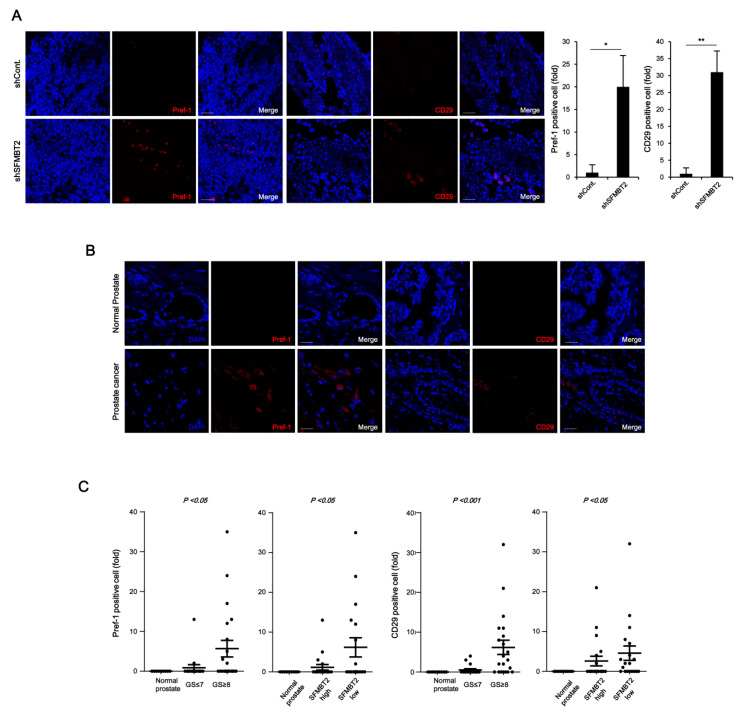
Down-regulation of SFMBT2 induces an increased number of preadipocytes and tumor-associated macrophages (TAMs) in prostate cancer. (**A**) Increased number of preadipocytes in prostate cancer induced by intraprostatic injection of LNCaP cells stably transfected with SFMBT2 shRNA. LNCaP cells (1 × 10^6^ cells) transfected stably with control (shCont) or SFMBT2 shRNA (shSFMBT2) were injected into the dorsal lobe of the prostate of nude mouse (n = 3/group). At week 5 post-injection, the prostate was harvested, fixed, sectioned, and immunostained with anti-Pref-1 and anti-CD29 antibodies. Representative images are shown. Nuclei were identified using DAPI staining. Scale bar, 25 μm. Pref-1 or CD29 positive cells were counted. (**B**) Increased expression of Pref-1 and anti-CD29 in prostate cancer tissues of patients. Immunohistochemical staining of a tissue array from prostate cancer patients was performed using anti-Pref-1 and anti-CD29 antibodies. Representative images are shown. Nuclei were identified using DAPI staining. Scale bar, 25 μm. (**C**) Proportional relationship of Gleason scores (GS) with the number of Pref-1 or CD29 positive preadipocytes in prostate cancer tissues of patients. Inverse relationship of SFMBT2 expression with a number of Pref-1 or CD29 positive preadipocytes in prostate cancer tissues of patients. (**D**) Increased number of TAMs in prostate cancer induced by intraprostatic injection of LNCaP cells stably transfected with SFMBT2 shRNA. LNCaP cells (1 × 10^6^ cells) transfected stably with control or SFMBT2 shRNA were injected into the dorsal lobe of the prostate of nude mice (n = 3/group). At week 5 post-injection, the prostate was harvested, fixed, sectioned, and immunostained with anti-CD163 and anti-CD206 antibodies. Representative images are shown. Nuclei were identified using DAPI staining. Scale bar, 25 μm. CD163 or CD206 positive cells were counted. (**E**) Increased expression of CD163 and CD206 in prostate cancer tissues of patients. Immunohistochemical staining of a tissue array from prostate cancer patients was performed using anti-CD163 and anti-CD206 antibodies. Representative images are shown. Nuclei were identified using DAPI staining. Scale bar, 25 μm. (**F**) Proportional relationship of Gleason scores with the number of CD163 and CD206 positive TAMs in prostate cancer tissues of patients. Inverse relationship of SFMBT2 expression with a number of CD163 and CD206 positive TAMs in prostate cancer tissues of patients. All data represent mean ± S.E.M. Significance values were * *p* ≤ 0.05 and ** *p* ≤ 0.01.

**Figure 3 cancers-12-02718-f003:**
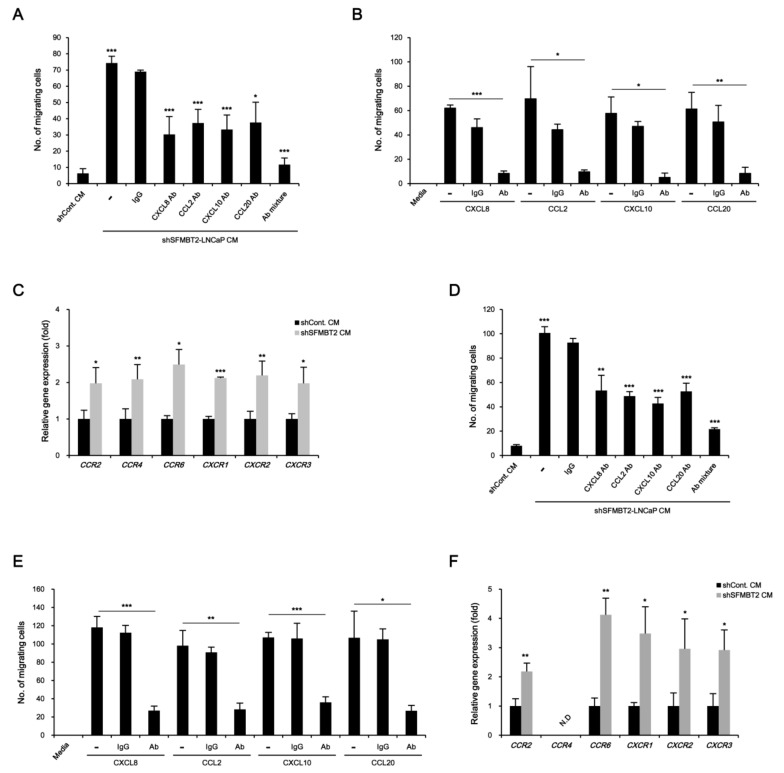
Chemokine-mediated migration of preadipocytes and TAMs. (**A**) Culture media (CM) from LNCaP cells stably transfected with SFMBT2 shRNA promote the migration of 3T3-L1 preadipocytes. However, the addition of antibodies against CXCL8, CCL2, CXCL10, and CCL20, and a mixture of antibodies reduces the migration of 3T3-L1 preadipocytes. After incubation of 3T3-L1 preadipocytes with antibodies against indicated chemokines in culture media from LNCaP cells stably transfected with SFMBT2 (shSFMBT2) or control shRNA (shCont), transwell assay was performed. Migrating 3T3-L1 preadipocytes were counted. (**B**) Treatment of CXCL8, CCL2, CXCL10, and CCL20 promotes the migration of 3T3-L1 preadipocytes. The addition of antibodies against CXCL8, CCL2, CXCL10, and CCL20 reduces the migration of 3T3-L1 preadipocytes. After incubation of 3T3-L1 preadipocytes with antibodies against indicated chemokines in RPMI 1640 media, transwell assay was performed. Migrating 3T3-L1 preadipocytes were counted. (**C**) Increased expression of CCR2, CCR4, CCR6, CXCR1, CXCR2, and CXCR3 in 3T3-L1 preadipocytes incubated with culture media from LNCaP cells stably transfected with SFMBT2 shRNA. RNA was analyzed. (**D**) Culture media from LNCaP cells stably transfected with SFMBT2 shRNA promote the migration of TAMs. However, the addition of antibodies against CXCL8, CCL2, CXCL10, and CCL20, and a mixture of antibodies reduces the migration of TAMs. After incubation of TAMs polarized from Raw264.7 cells with antibodies against indicated chemokines in culture media from LNCaP cells stably transfected with SFMBT2 or control shRNA, transwell assay was performed. Migrating TAMs were counted. (**E**) Treatment of CXCL8, CCL2, CXCL10, and CCL20 promotes the migration of TAMs. However, the addition of antibodies against CXCL8, CCL2, CXCL10, and CCL20 reduces the migration of TAMs. After incubation of TAMs polarized from Raw264.7 cells with antibodies against indicated chemokines in RPMI 1640 media, transwell assay was performed. Migrating TAMs were counted. (**F**) Increased expression of CCR2, CCR6, CXCR1, CXCR2, and CXCR3 in TAMs polarized from Raw264.7 cells incubated with culture media from LNCaP cells stably transfected with SFMBT2 shRNA. RNA was analyzed. All data represent mean ± S.E.M. Significance values were * *p* ≤ 0.05, ** *p* ≤ 0.01, and *** *p* ≤ 0.005.

**Figure 4 cancers-12-02718-f004:**
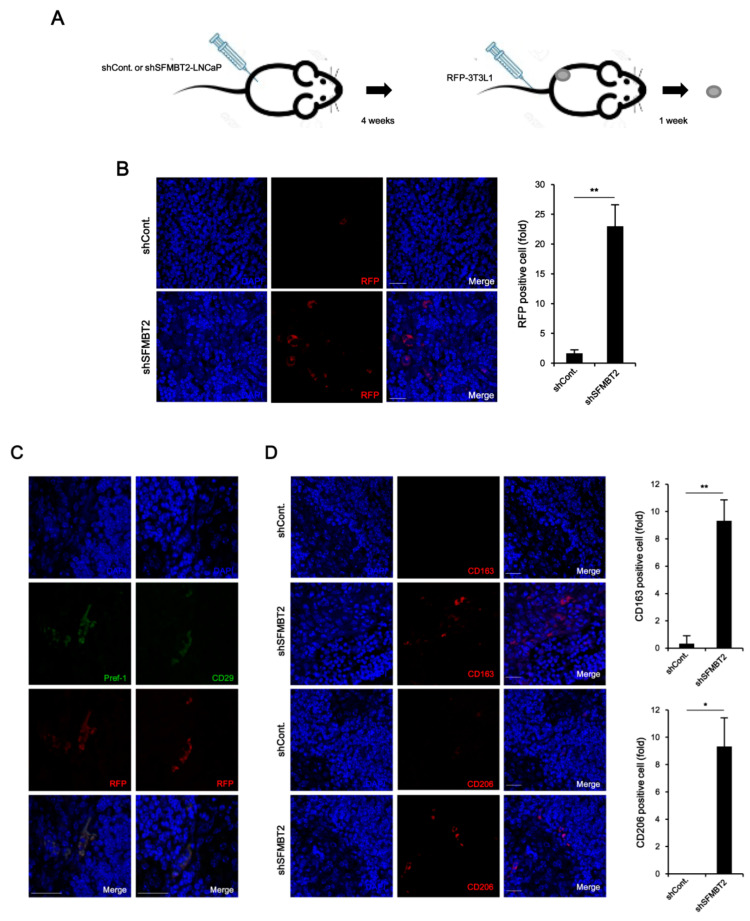
SFMBT2 knockdown LNCaP cells induce the infiltration of preadipocytes and TAMs in the xenograft model. (**A**) Schematic diagram of the xenograft experiment. LNCaP cells stably transfected with control (shCont) or SFMBT2 shRNA (shSFMBT2) were implanted subcutaneously into the flank of nude mice. After 4 weeks, stably RFP-expressing 3T3-L1 cells were injected into the tail vein of the same mouse. One week later, the tumor was harvested, fixed, sectioned, and immunostained with anti-RFP, anti-Pref-1, and anti-CD29 antibodies. (**B**) Increased number of infiltrated RFP-expressing 3T3-L1 preadipocytes in tumors. Representative images are shown. Nuclei were identified using DAPI staining. Scale bar, 25 μm. RFP positive cells were counted. (**C**) RFP expressing cells are Pref-1 and CD29 positive in tumors. Representative images are shown. Scale bar, 25 μm. (**D**) Increased number of infiltrated TAMs immunostained with anti-CD163 and anti-CD206 antibodies in tumors. Representative images are shown. Nuclei were identified using DAPI staining. Scale bar, 25 μm. CD163 or CD206 positive cells were counted. All data represent mean ± S.E.M. Significance values were * *p* ≤ 0.05 and ** *p* ≤ 0.01.

**Figure 5 cancers-12-02718-f005:**
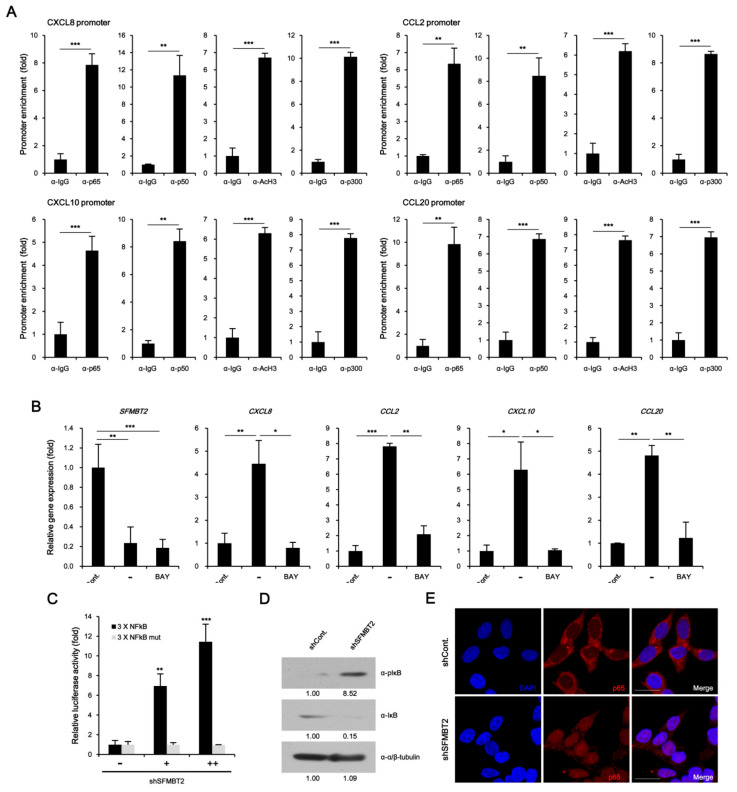
Activation of NF-κB results in the up-regulation of chemokine gene expression in STMBT2 knockdown LNCaP cells. (**A**) Enrichment of NF-κB (p50, p65), p300, and acetylated histone H3 to CXCL8, CCL2, CXCL10, and CCL20 gene promoters in LNCaP cells stably transfected with SFMBT2 shRNA. ChIP assay was performed using anti-NF-κB p50, anti-NF-κB p65, anti-p300, and anti-acetylated histone H3 (AcH3) antibodies. The occupancy of each protein was determined with quantitative PCR in gene promoters encompassing the NF-κB binding sites. ChIP was performed using normal IgG as a control. (**B**) NF-κB activation is required for the up-regulation of CXCL8, CCL2, CXCL10, and CCL20 gene expression in LNCaP cells stably transfected with SFMBT2 shRNA. RNA was analyzed in LNCaP cells stably transfected with control (shCont) or SFMBT2 shRNA (shSFMBT2) with or without BAY 11–7085 treatment. (**C**) Increased activity of reporter containing wild-type NF-κB binding sites in LNCaP cells stably transfected with SFMBT2 shRNA. However, reporter containing mutated NF-κB binding sites was not activated in LNCaP cells stably transfected with SFMBT2 shRNA. (**D**) Increased phosphorylation and degradation of IκB in LNCaP cells stably transfected with SFMBT2 shRNA. Lysates from LNCaP cells stably transfected with control or SFMBT2 shRNA were immunoblotted with anti-IκB and anti-phospho IκB antibodies. The anti-tubulin antibody was used for loading control. Western blots were analyzed quantitatively using ImageJ program. (**E**) Increased nuclear translocation of NF-κB p65 in LNCaP cells stably transfected with SFMBT2 shRNA. Cells were immunostained with anti-NF-κB p65 antibody. Representative images are shown. Nuclei were identified using DAPI staining. Scale bar, 25 μm. All data represent mean ± S.E.M. Significance values were * *p* ≤ 0.05, ** *p* ≤ 0.01, and *** *p* ≤ 0.005.

**Figure 6 cancers-12-02718-f006:**
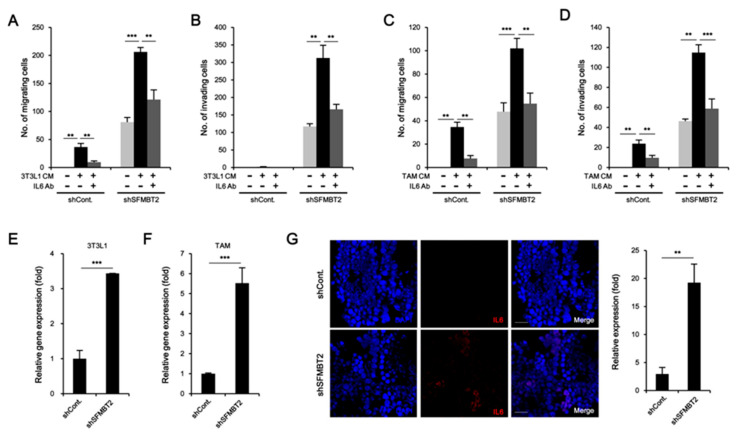
Preadipocytes and TAMs promote migration and invasion of prostate cancer cells through the up-regulation of IL-6. (**A**,**B**) Culture media (CM) from 3T3-L1 preadipocytes promote migration and invasion of LNCaP cells stably transfected with SFMBT2 shRNA. However, the addition of anti-IL-6 antibody suppresses migration and invasion of LNCaP cells stably transfected with SFMBT2 shRNA. After incubation of LNCaP cells stably transfected with SFMBT2 (shSFMBT2) or control shRNA (shCont) with anti-IL-6 antibody in culture media from 3T3-L1 preadipocytes, transwell assay was performed. Migrating or invading LNCaP cells were counted. (**C**,**D**) Culture media from TAMs polarized from Raw264.7 cells promote increased migration and invasion of LNCaP cells stably transfected with SFMBT2 shRNA. However, the addition of the anti-IL-6 antibody suppresses migration and invasion of LNCaP cells stably transfected with SFMBT2 shRNA. After incubation of LNCaP cells stably transfected with SFMBT2 (shSFMBT2) or control shRNA (shCont) with anti-IL-6 antibody in culture media from TAMs, transwell assay was performed. Migrating or invading LNCaP cells were counted. (**E**,**F**) Increased expression of IL-6 in 3T3-L1 preadipocytes and TAMs polarized from Raw264.7 cells incubated with culture media from LNCaP cells stably transfected with SFMBT2 shRNA. RNA was analyzed. (**G**) Increased expression of IL-6 in prostate cancer induced by intraprostatic injection of LNCaP cells stably transfected with SFMBT2 shRNA. LNCaP cells (1 × 10^6^ cells) stably transfected with control or SFMBT2 shRNA were injected into the prostate (dorsal lobe) of nude mice (n = 3/group). At week 5 post-injection, the prostate was harvested, fixed, sectioned, and immunostained with the anti-IL-6 antibody. Representative images are shown. Nuclei were identified using DAPI staining. Scale bar, 25 μm. Expression was quantified using the ImageJ program. (**H**) Increased expression of IL-6 in prostate cancer tissues of patients. Immunohistochemical staining of a tissue array from prostate cancer patients was performed using the anti-IL-6 antibody. Representative images are shown. Nuclei were identified using DAPI staining. Scale bar, 25 μm. Expression was quantified using the ImageJ program. (**I**) Proportional relationship of Gleason scores (GS) with the expression level of IL-6 in prostate cancer tissues of patients. Inverse relationship of SFMBT2 expression with the expression level of IL-6 in prostate cancer tissues of patients. All data represent mean ± S.E.M. Significance values were ** *p* ≤ 0.01 and *** *p* ≤ 0.005.

**Figure 7 cancers-12-02718-f007:**
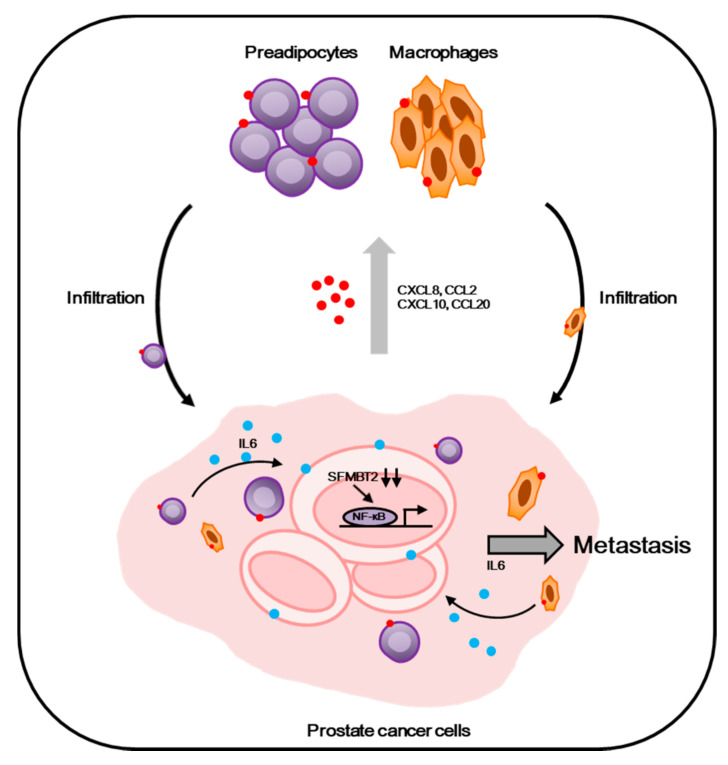
Proposed model. Prostate cancer cells expressing a low level of SFMBT2, which is correlated with a higher Gleason score (≥8), activate NF-κB signaling and up-regulate the expression of chemokines such as CXCL8, CCL2, CXCL10, and CCL20. This event may promote the infiltration of preadipocytes and TAMs into the tumor microenvironment. In addition, IL-6 is up-regulated in infiltrated preadipocytes and TAMs, resulting in enhanced migration and invasion of prostate cancer cells for metastasis.

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
