# Peer review of "SFMBT2-Mediated Infiltration of Preadipocytes and TAMs in Prostate Cancer"

_cancers, 2020, doi:10.3390/cancers12092718_

Round 1
Reviewer 1 Report
A nice, stringent and interesting elucidation of cytokine-derived crosstalk between prostate cancer cells and invading cells, which appears to be regulated by SFMBT2 in cancer cells. The authors identified a mixture of the cytokines CXCL8 (IL-8), CCL2 (MCP-1), CXCL10 and CCL20 which are produced by the prostate cancer cells in the absence of SFMBT2 that act on murine pre-adipocytes and TAMs to induce proliferation, migration and invasion in vitro and in mouse xenografts. SFMBT2 was not shown to bind to promoters of the respective cytokines, but appears to regulate their expression via the NFkB pathway. In return, cytokines secreted from SFMBT2-depleted prostate carcinoma cells elicit a secretory response in the form of IL-6, and this cytokine is also found to be elevated in xenografts and human prostate cancer tissues.
Major Concerns:
- Apart from the well-known invasion of TAMs, the authors present pre-adipocytes another cell type capable of invasion into prostate carcinomas. The identification of this novel cell type is based on the marker Pref-1 (DLK2), while the second marker used in this study, CD29 (b1 integrin) would only help differentiate between pre-adipocytes and mature adipocytes. A second independent marker for pre-adipocytes would be favorable to inequivocally identify this cell type in the context of invasion into prostate cancer.
- In contrast to the human androgen-dependent prostate cancer cell line LNCaP, murine cell lines (pre-adipocytes 3T3-L1 and monocytic Raw264.7) were used for the analysis of secreted cytokines regulated by SFMBT2. Taking the species interplay in mouse xenografts injected with human prostate cancer cells into account, these cytokines may have different activities in a human cancer to a human target cell situation. This should be discussed in more detail.
- Moreover, a more detailed analysis to identify the cytokine that predominates in exerting the described effects would be desirable.
- The authors published SFMBT2 previously to act on the promoter of HOXB13 in conjunction with YY1 in the AR-negative prostate cancer cell line DU145. However, a possible role of HOXB13, a known player in prostate cancer progression, as activator in the context of prostate cancer cytokine production was not discussed and should be included.
- Unfortunately, the quality of the figures is extremely low, such that immunofluorescence data would not support the description of the data presents in the results part.
Altogether, with minor criticisms, we would recommend this story for publication.
Author Response
A nice, stringent and interesting elucidation of cytokine-derived crosstalk between prostate cancer cells and invading cells, which appears to be regulated by SFMBT2 in cancer cells. The authors identified a mixture of the cytokines CXCL8 (IL-8), CCL2 (MCP-1), CXCL10 and CCL20 which are produced by the prostate cancer cells in the absence of SFMBT2 that act on murine pre-adipocytes and TAMs to induce proliferation, migration and invasion in vitro and in mouse xenografts. SFMBT2 was not shown to bind to promoters of the respective cytokines, but appears to regulate their expression via the NFkB pathway. In return, cytokines secreted from SFMBT2-depleted prostate carcinoma cells elicit a secretory response in the form of IL-6, and this cytokine is also found to be elevated in xenografts and human prostate cancer tissues.
; We appreciate the Reviewer #1 for the valuable comments. We performed additional experiments as suggested. We highlighted changes in red in our revised manuscript. We believe that our manuscript is greatly modified and improved.
Major Concerns:
Apart from the well-known invasion of TAMs, the authors present pre-adipocytes another cell type capable of invasion into prostate carcinomas. The identification of this novel cell type is based on the marker Pref-1 (DLK2), while the second marker used in this study, CD29 (b1 integrin) would only help differentiate between pre-adipocytes and mature adipocytes. A second independent marker for pre-adipocytes would be favorable to inequivocally identify this cell type in the context of invasion into prostate cancer.
; Because it takes a month to get antibodies against second marker for pre-adipocyte from vendors and deadline (7 August) for resubmission of the revised manuscript, we could not perform immunohistochemistry. Instead, we performed RT-PCR to examine expression level of additional marker genes such as CD24, Sca-1, C/EBPα, and FABP4 in pre and mature adipocytes. Consistently, we found increased expression of marker genes for preadipocytes (CD24 and Sca-1) and unchanged or slightly increased expression of marker genes for mature adipocytes (C/EBPα and FABP4). We placed the data in Supplementary Fig S3.
In contrast to the human androgen-dependent prostate cancer cell line LNCaP, murine cell lines (pre-adipocytes 3T3-L1 and monocytic Raw264.7) were used for the analysis of secreted cytokines regulated by SFMBT2. Taking the species interplay in mouse xenografts injected with human prostate cancer cells into account, these cytokines may have different activities in a human cancer to a human target cell situation. This should be discussed in more detail.
; As Reviewer #1 suggested, we described different activities of cytokines induced by SFMBT2 knockdown in different cell types of prostate cancer tissue in the Discussion.
Moreover, a more detailed analysis to identify the cytokine that predominates in exerting the described effects would be desirable.
; We agreed Reviwer#1’s comment. Due to deadline of resubmission, we could not perform additional experiments at present. However, we are going to investigate systemic analysis for cytokine expression using RNA seq. or array for further study.
The authors published SFMBT2 previously to act on the promoter of HOXB13 in conjunction with YY1 in the AR-negative prostate cancer cell line DU145. However, a possible role of HOXB13, a known player in prostate cancer progression, as activator in the context of prostate cancer cytokine production was not discussed and should be included.
; We appreciate the Reviewer #1 for the insightful comment. As suggested, we described a possible role of HoxB13 in prostate cancer metastasis in the Discussion and cited two references.
Unfortunately, the quality of the figures is extremely low, such that immunofluorescence data would not support the description of the data presents in the results part.
; We apology for low quality of figures. We previously used Adobe Photoshop the figure construction. In this revised manuscript, we used Adobe Illustrator for better quality of figures with high resolution.
Altogether, with minor criticisms, we would recommend this story for publication.
; We appreciate Reviewer #1 for the valuable comments.
Reviewer 2 Report
Jungsug Gwak et al. explore the role of SFMBT2 in prostate cancer progression/metastasis, building upon the group's prior publications in this field. They observe that aggressive prostate cancers have low expression of SFMBT2 and high IL6. They create an inducible knockdown model to recapitulate downregulation of SFMBT2 and observe upregulation of multiple chemokines/cytokines and recruitment of preadipocytes in vivo. Cellular experiments with the chemokines themselves vs anti-chemokine antibodies produced similar results to regulation of SFMBT2. The proposed mechanism of regulation of the chemokines by SFMBT2 was through NFkB. IL6 secreted from preadipocytes was also shown to be associated with low SFMBT2 and invasion/migration of prostate cancer cells. Overall the study is presented logically with clear rationale, methods, and conclusions throughout. Overall I believe the work is high quality.
In terms of clear presentation of the results, I would ensure that the presented figures are adequately readable in terms or text/resolution of figures, and of adequate resolution. The provided proof had borderline image and text quality for the presented figures.
Author Response
Review report 2
Jungsug Gwak et al. explore the role of SFMBT2 in prostate cancer progression/metastasis, building upon the group's prior publications in this field. They observe that aggressive prostate cancers have low expression of SFMBT2 and high IL6. They create an inducible knockdown model to recapitulate downregulation of SFMBT2 and observe upregulation of multiple chemokines/cytokines and recruitment of preadipocytes in vivo. Cellular experiments with the chemokines themselves vs anti-chemokine antibodies produced similar results to regulation of SFMBT2. The proposed mechanism of regulation of the chemokines by SFMBT2 was through NFkB. IL6 secreted from preadipocytes was also shown to be associated with low SFMBT2 and invasion/migration of prostate cancer cells. Overall the study is presented logically with clear rationale, methods, and conclusions throughout. Overall I believe the work is high quality.
; We appreciate the Reviewer #2 for the valuable comments. We performed additional experiments as suggested. We highlighted changes in red in our revised manuscript. We believe that our manuscript is greatly modified and improved.
In terms of clear presentation of the results, I would ensure that the presented figures are adequately readable in terms or text/resolution of figures, and of adequate resolution. The provided proof had borderline image and text quality for the presented figures.
; For the better images in the revised manuscript, we used Adobe Illustrator for the figure construction. In addition, the manuscript was re-edited by native speaker. We appreciate Reviewer #2 for the valuable comment.
Round 2
Reviewer 1 Report
All our comments were addressed and we recommend to accept the paper in the current form.